# TROAP Promotes the Proliferation, Migration, and Metastasis of Kidney Renal Clear Cell Carcinoma with the Help of STAT3

**DOI:** 10.3390/ijms24119658

**Published:** 2023-06-02

**Authors:** Jun Wang, Hongyuan Wan, Yuanyuan Mi, Sheng Wu, Jie Li, Lijie Zhu

**Affiliations:** 1Department of Urology, Affiliated Hospital of Jiangnan University, Wuxi 214122, Chinaminiao1984@163.com (Y.M.);; 2Department of Urology, First Affiliated Hospital of Nanjing Medical University, Nanjing 210008, China; 3Wuxi Medical College, Jiangnan University, Wuxi 214122, China

**Keywords:** kidney renal clear cell carcinoma, trophinin-associated protein, STAT3, proliferation, migration

## Abstract

Kidney renal clear cell carcinoma (KIRC) is a subtype of renal cell carcinoma that threatens human health. The mechanism by which the trophinin-associated protein (TROAP)–an important oncogenic factor–functions in KIRC has not been studied. This study investigated the specific mechanism by which TROAP functions in KIRC. *TROAP* expression in KIRC was analyzed using the RNAseq dataset from the Cancer Genome Atlas (TCGA) online database. The Mann–Whitney U test was used to analyze the expression of this gene from clinical data. The Kaplan–Meier method was used for the survival analysis of KIRC. The expression level of *TROAP* mRNA in the cells was detected using qRT-PCR. The proliferation, migration, apoptosis, and cell cycle of KIRC were detected using Celigo, MTT, wound healing, cell invasion assay, and flow cytometry. A mouse subcutaneous xenograft experiment was designed to demonstrate the effect of *TROAP* expression on KIRC growth in vivo. To further investigate the regulatory mechanism of TROAP, we performed co-immunoprecipitation (CO-IP) and shotgun liquid chromatography–tandem mass spectrometry (LC-MS). TCGA-related bioinformatics analysis showed that *TROAP* was significantly overexpressed in KIRC tissues and was related to higher T and pathological stages, and a poor prognosis. The inhibition of *TROAP* expression significantly reduced the proliferation of KIRC, affected the cell cycle, promoted cell apoptosis, and reduced cell migration and invasion. The subcutaneous xenograft experiments showed that the size and weight of the tumors in mice were significantly reduced after *TROAP*-knockdown. CO-IP and post-mass spectrometry bioinformatics analyses revealed that TROAP may combine with signal transducer and activator of transcription 3 (STAT3) to achieve tumor progression in KIRC; this was verified by functional recovery experiments. TROAP may regulate KIRC proliferation, migration, and metastasis by binding to STAT3.

## 1. Introduction

The incidence of kidney cancer was ranked 16th among the 36 major cancers globally in 2020, with 431,288 new cases and 179,368 deaths [1]. There were 75,800 new cancer cases in China in 2016, including 48,000 male and 27,800 female cases [2]. The reasons for the higher incidence in developed countries are unknown; however, it may be linked to genomic, occupational, and other environmental exposures, such as smoking [3]. Kidney cancer poses a serious threat to human health.

Kidney renal clear cell carcinoma (KIRC) is the most common subtype of renal cell carcinoma (RCC), accounting for approximately 70% of cases. They are adenocarcinomas derived from renal tubular epithelial cells and are relatively rich in glycogen, phosphoesters, neutral fats, and other substances. Histologically, they are surrounded by a clear cytoplasm, nested cell clusters, and dense endothelial networks [3]. Although surgical resection is the primary treatment for KIRC, there is still a recurrence rate of approximately 30% and the possibility of distant metastasis. Additionally, the 5-year survival rate after metastasis is low, and conventional chemotherapy and radiotherapy are ineffective in treating all RCC subtypes. Therefore, new treatments, especially molecular targeted therapy, are the main focus of KIRC research [4,5,6,7].

Trophinin-associated protein (TROAP) is a cytoplasmic protein consisting of 778 amino acid residues rich in proline, and contains three domains homologous to src. The human *TROAP* gene encodes the TROAP protein localized on chromosome 12q13.12 [8,9,10]. TROAP is related to bystin and trophinin, which are involved in the initial adhesion process between blastocysts and uterine epithelial cells, and may form complexes involved in embryo implantation [11]. Multiple studies have shown that TROAP expression in breast and colorectal cancer, ovarian adenocarcinoma, hepatocellular carcinoma, gastric cancer and other tumors enhances malignancy and promotes tumor development [10,12,13,14,15]. Additionally, data have shown that elevated *TROAP* expression positively correlates with the clinical severity of hepatocellular carcinoma, indicating a poor overall and disease-free survival [10]. TROAP can affect tumor progression through various pathways; for example, it can promote disease progression through the Wnt3/survival protein signaling pathway in prostate cancer [16]. In addition, Han Liu et al. found that TROAP regulated cell division cycle 20 (CDC20) and influenced spindle microtubules (ASPM) to promote malignant tumor development during the S and G2 phases of the cell cycle [17]. Lastly, *TROAP* can be used as the target of some microRNAs to inhibit the proliferative activity of cancer cells [13]. For instance, in KIRC, miR-532-3p reduced cancer cell viability, migration, and invasion by targeting *TROAP* [18]. However, the downstream regulatory mechanism of TROAP in KIRC has not been elucidated.

In this study, we defined the expression of *TROAP* in KIRC cells, and explored the role and downstream mechanism of TROAP in KIRC cells.

## 2. Results

### 2.1. TROAP Is Highly Expressed in KIRC

All differentially expressed genes were identified from the TCGA database, and the results are shown in Figure 1a. In the TCGA database, the expression of TROAP in cancer tissue samples was significantly higher than that in the adjacent tissues (Figure 1b,c). The Mann–Whitney U test was used to analyze the significance of the expression of this gene at different levels across different clinical data. TROAP expression in cancer tissues of patients with different T and pathological stages of KIRC was significantly different (Table 1), suggesting a correlation (Figure 1d). We tested the TROAP expression using the Kaplan–Meier method (log-rank test statistic). We found that it significantly impacted survival, and patients with higher TROAP expression had a worse prognosis (Figure 1e), indicating that TROAP expression was upregulated in KIRC, contributing to the poor prognosis of KIRC patients.

Therefore, to verify the differential upregulation of *TROAP* expression in KIRC, we determined the *TROAP* mRNA expression levels in the following KIRC cell lines: 786-O, ACHN, and Caki-1 cell lines. *TROAP* mRNA expression was high in the KIRC cell lines (786-O, ACHN, and Caki-1) (Figure 1f). Combined with our bioinformatics analysis of *TROAP* expression in KIRC and experimental detection of *TROAP* expression in cell lines, we confirmed that *TROAP* was highly expressed in KIRC.

### 2.2. TROAP Promoted KIRC Cell Proliferation, Migration, and Invasion, and Affected the Cell Cycle and Apoptosis

After 3 days of shRNA lentivirus infection, the mRNA and protein expression of TROAP in 786-O and ACHN cells in the experimental group were inhibited (Figure 2a,b). The Celigo and MTT assays revealed that 786-O and ACHN cell proliferation rates were significantly inhibited after *TROAP*-knockdown. These results suggest that TROAP significantly affected the 786-O and ACHN cell proliferation (Figure 2c and Appendix A). The results of the Celigo wound-healing and migration assays suggest that TROAP was significantly correlated with the migratory ability of 786-O and ACHN cells (Figure 2d). The invasion (transwell) assay showed that the invasion and metastasis ability of 786-O and ACHN cells in the experimental group was significantly inhibited after 3 days of shRNA lentivirus infection, suggesting a significant correlation with TROAP (Figure 2e and Appendix A). After *TROAP* knockdown, the number of S-phase cells in the 786-O cell group decreased (*p* < 0.05), ACHN cells decreased in the S phase (*p* < 0.05), the number of cells in the G1 phase increased (*p* < 0.05), and the number of cells in the G2/M phase decreased (*p* < 0.05). These results suggest that the *TROAP* gene was significantly related to the 786-O and ACHN cell cycles (Figure 2f). Furthermore, 786-O and ACHN cell apoptosis was significantly increased after transfection with shTROAP compared with the shCtrl group, suggesting a significant correlation with TROAP (Figure 2g). Combined with the above results, we confirmed that the TROAP interference could inhibit the 786-O and ACHN cell proliferation, migration, and invasion; additionally, it could reverse the effect on the cell cycle and apoptosis of these cells.

### 2.3. TROAP-Knockdown Inhibited KIRC Growth In Vivo

As shown in Figure 3a, tumors in mice injected with shTROAP-transduced cells were significantly smaller than those in mice injected with shCtrl cells. Compared with the shCtrl group, the volume and weight of the tumors derived from shTROAP KIRC cells were significantly reduced (Figure 3b,c), suggesting that *TROAP* downregulation inhibits KIRC tumor growth in vivo.

### 2.4. TROAP Functions in KIRC Cells by Binding to STAT3

We obtained 201 candidate interacting proteins for biosignal analysis and plotted the gene interaction network based on these results (Appendix A). In order to study the regulatory mechanism between TROAP and downstream proteins in KIRC cells, 3× FLAG–TROAP fusion lentivirus was constructed to infect the 786-O cells. FLAG beads (Sigma) were used for CO-IP. The protein complexes were purified and the CO-IP samples were subjected to SDS-PAGE and Coomassie brilliant blue staining (Figure 4a). Four top candidate proteins were selected for Western blot verification, including STAT3, PLK1, RB1, and NPM1. As shown in Figure 4b, FLAG was detected in all OE groups with *TROAP* overexpression, indicating that the IP process was successful. Notably, TROAP could directly bind to the activated p-STAT3 protein, while it could not bind to p-PLK1, p-RB1, and p-NPM1. We also found that p-STAT3 protein levels were notably elevated by *TROAP* overexpression, and reduced with *TROAP* knockdown (Figure 4b,c). Subcellular analysis confirmed that *TROAP* knockout significantly reduced the nuclear enrichment of p-STAT3 protein, in contrast to no differences observed in the cytoplasm (Figure 4d). In subcutaneous models, the levels of p-STAT3 were found to be significantly reduced in *TROAP*-knockdown tumors compared with control tumors, while other proteins, including p-PLK1, p-RB1, and p-NPM1, were not changed obviously (Figure 4e).

### 2.5. STAT3 Overexpression Restored the Decreased Proliferation and Metastasis of KIRC Induced by TROAP Interference

Next, we determined the reliability of the protein interactions in KIRC. We verified the functional recovery effect of STAT3 on TROAP in vitro. The *TROAP* lentivirus interference and *STAT3* lentivirus overexpression in 786-O cells were constructed and divided into KD and OE groups, respectively, which were confirmed by qPCR (Figure 5a). Cell growth was monitored using Celigo, and the cells infected with an empty viral vector were used as control (NC). The results showed that KD + NC group proliferation was significantly slower than that of the NC + NC group, as we expected. Compared with the KD + NC group, the slowing trend of proliferation was significantly restored in KD + OE (Figure 5b). The MTT assay was subsequently used to detect the changes in cell viability in the three groups to verify the functional recovery effect of the *STAT3* gene on the *TROAP* gene. We found that compared with the NC + NC group, the cell viability of the KD + NC group decreased, as expected; compared with the KD + NC group, the cell viability of the KD + OE group increased. The results indicate that STAT3 gene overexpression restored the proliferative function of the *TROAP*-knockdown group (Figure 5c). The trans-well assay showed that compared with the NC + NC group, the metastatic ability of the KD + NC group was weakened, consistent with the expected result. However, the metastatic ability of the KD + OE group was enhanced compared with the KD + NC group (Figure 5d). This indicated that STAT3 overexpression could restore the metastatic function of *TROAP*-knockdown cells. We concluded that STAT3 overexpression restored the decreased proliferation and metastatic KIRC by *TROAP* overexpression.

## 3. Discussion

In this study, through bioinformatics analysis and the TCGA database, we screened out a significantly different *TROAP* gene, which had a higher expression level in KIRC cancer tissues than in adjacent tissues. Through test analysis, we found that the high expression of TROAP was related to the T and pathological stages of KIRC. The upregulation of TROAP may be one of the factors leading to a poor prognosis in patients with KIRC. Similarly, TROAP is upregulated in other malignant tumors, such as liver, ovarian, prostate, and colon cancers [10,12,13,19]. The high expressions of *TROAP* could promote the proliferation and metastasis of cancer cells, leading to a poor prognosis [14]. Therefore, we performed qPCR on KIRC cell lines (786-O, ACHN, and CAKI-1) and found that *TROAP* mRNA was highly expressed in these cell lines. This proves that *TROAP* expression was upregulated in KIRC. Subsequently, we used lentivirus to interfere with *TROAP* expression in 786-O and ACHN cells. We observed that TROAP promoted KIRC cell proliferation, migration, and invasion, and affected the cell cycle and apoptosis. We then considered whether interfering with TROAP could affect tumor growth in vivo. Xenograft tumor experiments in nude mice confirmed our suspicion that tumor growth was slowed in the *TROAP* knockdown group. These experiments demonstrated that TROAP is an oncogene that could aggravate the malignancy of cancer cells.

Additionally, we explored the specific regulatory mechanism by which TROAP exerts its oncogenic effects in KIRC. Mir-532-3p, upstream of *TROAP*, reportedly attenuates KIRC cell viability, migration, and invasion [18]. However, its mechanism of action was not explored; we performed CO-IP experiments to achieve this. After mass spectrometry analysis, screening, and Western blot experiments, we found that TROAP can combine with STAT3. TROAP could directly bind to the activated p-STAT3 protein; the p-STAT3 protein levels were notably elevated by *TROAP* overexpression and reduced with *TROAP*-knockdown. *TROAP*-knockdown remarkably decreased the nuclear enrichment of p-STAT3 protein. In subcutaneous models, p-STAT3 levels were found to be significantly lower in *TROAP*-knockout tumors compared to control tumors, which is consistent with in vitro findings. STAT3 overexpression prevented inhibition of KIRC proliferation and metastasis after *TROAP* interference. The combined effects of TROAP and STAT3 may contribute to the carcinogenic effect.

STAT protein family is a family of cytoplasmic transcription factors, including STAT3 [20]. STAT3 consists of 770 amino acids and contains 6 functionally conserved domains. As a cytokine receptor, STAT3 is tightly regulated by negative modulators and is inactive when unstimulated. Upon stimulation (primarily through phosphorylation induced by upstream ligands), STAT3 dimerizes and translocates to the nucleus, where it binds to DNA and becomes functional [21]. As a member of the cytokine receptor family, STAT3 plays an important role in the proliferation and metastasis of various cancers. For example, STAT3 participates in the regulation of SOX4 to promote tumor progression in hepatic cell carcinoma [22]. The IL6/STAT3 pathway hijacked estrogen receptor alpha enhancers to promote cancer cell proliferation in breast cancer [23]. Similarly, we found that in KIRC, STAT3 promoted tumor proliferation, invasion, and metastasis, and played a prooncogenic role in KIRC.

TROAP is reportedly involved in the initial adhesion process between blastocysts and uterine epithelial cells. Additionally, it is associated with the microscopic cytoskeleton and mitosis and could affect the cell cycle [8,24]. Some pathways involved in TROAP have been identified in prostate, liver, and colorectal cancers [13,16,25]. We found that STAT3 could act as a TROAP-interacting protein and that TROAP could promote KIRC proliferation, migration, and metastasis with the help of STAT3. In 786-O cells, STAT3 overexpression restored the decreased cell proliferation and metastasis caused by *TROAP*-knockdown, indicating that TROAP can bind to STAT3 to regulate the degree of KIRC malignancy. In our research, we investigated the specific binding mode of TROAP and STAT3, and verified whether the binding of TROAP and STAT3 affects the degree of STAT3 phosphorylation. However, we did not confirm the correlation between the TROAP and JAK/STAT3 pathways in KIRC, which was the most studied pathway of them [26,27,28]. Furthermore, in this study, although we referred to and analyzed a large number of clinical samples of KIRC in public databases, and selected TROAP as the research target based on this, we did not collect clinical samples of KIRC patients for testing in subsequent studies, which may need to be gradually improved in future studies. In addition, we did not conduct caudal vein transfer experiments in nude mice to verify whether TROAP had the ability to promote KIRC distant transfer in vivo, which needs to be supplemented in our subsequent studies. In future, we will explore the effects of TROAP on the nuclear translocation of STAT3 and its binding to DNA. Further research on TROAP and STAT3 can provide a direction for selecting early diagnostic markers for KIRC and may bring new ideas for molecular targeted therapy development.

Overall, we concluded that TROAP was highly expressed in KIRC and promoted cancer cell proliferation, invasion, and metastasis. Additionally, the TROAP and STAT3 interaction affected KIRC proliferation and metastasis (Figure 6). However, a more specific mechanism needs to be explored.

## 4. Materials and Methods

### 4.1. Bioinformatics Analysis

*TROAP* expression was downloaded from the Cancer Genome Atlas (TCGA) online database (http://tcga-data.nci.nih.gov/tcga/) which were accessed on 25 June 2020. First, we selected keywords related to renal clear cell carcinoma on the data download page and downloaded the RNAseq files in batches. Based on the barcode information of the samples, the original data files of the 72 paired samples were split from the sample list (Appendix A). Second, for each gene symbol, we selected the transcript with the highest expression for analysis. If the number of raw reads of all samples corresponding to that symbol was <50, the data for that symbol was considered unavailable. The trimmed mean of the M-value (TMM) method was used for data normalization. Furthermore, we performed primary quality control by observing the biological coefficient of variation (BCV). A negative binomial general linear model was used for the statistical analyses of multiple paired samples to calculate the *p*-values with a filtering criterion of <0.05. Simultaneously, the fold-change (FC) value of the sample was calculated using log2 (Cancer/Normal) and the filtering standard as ≥1 and ≤−1. We used the Mann–Whitney U test to analyze the significance of the differences in *TROAP* gene expression levels at different levels of different clinical data. In this study, we used Spearman’s test to analyze the correlation between the *TROAP* expression level in cancer tissues and clinical data. We preliminarily explored the potential role of these clinical data in the occurrence and development of renal clear cell carcinoma. Lastly, the Kaplan–Meier method (log-rank test statistic) was used to test whether the effect of the *TROAP* gene expression level on survival time was significant. All TCGA data were calculated and processed using the R software (http://www.r-project.org) and SPSS 17.0.

### 4.2. Quantitative Real-Time PCR

RNA extraction from 786-O, ACHN, and Caki-1 cell lines was performed using the TRIzol kit (Pufei, Shanghai, China). RNA was then reverse-transcribed into cDNA using the M-MLV kit (Promega, Madison, MI, USA), according to the manufacturer’s instructions. The primer sequences used for GAPDH, TROAP, and STAT3 are listed in Appendix A. SYBR Master Mix (Takara, San Jose, CA, USA) was used for PCR on an ABI 7500 PCR system (ABI, Co. Ltd., Lees Summit, MO, USA). The qRT-PCR protocol included initiation at 95 °C for 15 s, followed by 45 cycles at 95 °C for 5 s and 60 °C for 30 s. The relative mRNA expression (TROAP/GAPDH or STAT3/GAPDH) was determined using the 2^−∆∆Ct^ method. All analyses were conducted in triplicate.

### 4.3. Cell Culture

The human KIRC cell lines 786-O, ACHN, and Caki-1 were purchased from the cell bank of the Chinese Academy of Science (Shanghai, China). The cells were cultured in Dulbecco modified Eagle’s medium (DMEM; Corning, NY, USA), supplemented with 10% fetal bovine serum (Ausbian, Adelaide, Australia). The incubator conditions were controlled at 37 °C and 5% CO_2_.

### 4.4. Lentivirus-Mediated Small Hairpin RNA Construction and Determination of Efficiency

ShTROAP was designed based on the *TROAP* (NM_005480, Infection https://www.ncbi.nlm.nih.gov/nuccore/NM_005480) target sequence (CCCCCGGCAAGCCACGAAGGATC), and a non-silencing shRNA sequence (TTCTCCGAACGTGTCACGT) was used as a negative control. After designing the RNA interference target, a single-stranded DNA oligo containing an interference sequence was synthesized and annealed to produce double-stranded DNA. Subsequently, the digested lentiviral vector was directly connected to the restriction sites at both ends. The ligation product was transferred into the prepared Escherichia coli competent cells, and the positive recombinant was identified via PCR and sent for sequencing verification. Subsequently, the sequencing results were compared with the correct clones for plasmid extraction. Additionally, the *TROAP* and *STAT3* overexpression lentiviruses, LV-TROAP and LV-STAT3, were constructed, and their sequences are listed in the table (Appendix A). The transfection efficiency was >80% based on green-fluorescent protein fluorescence.

### 4.5. Cell Proliferation and Colony Formation Assays

The proliferation of 786-O and ACHN cells was detected using Celigo and MTT assays. After reaching the logarithmic growth phase, both the shCtrl and shTROAP cell cultures were trypsinized and resuspended in DMEM. The cells were plated in five 96-well plates, followed by incubation at 37 °C and 5% CO_2_. After incubation for 1, 2, 3, 4, and 5 days, the cells were incubated with 20 μL of 5 mg/mL MTT for 4 h. Lastly, we measured the optical density at a wavelength of 490 nm. Based on these data, we performed statistical data mapping and constructed cell proliferation curves. The assay was performed in triplicate. In the Celigo assay, cell collection and plate-laying were the same as those in the MTT assay. Celigo was performed 1, 2, 3, 4, and 5 days after plate-laying. The number of cells with green fluorescence in the plates of each scan well was accurately calculated by adjusting the input parameters of the analysis settings. The data were statistically plotted, and the cell proliferation curves were planned for the 5 days.

### 4.6. Apoptosis Analysis and Cell Cycle Detection

Flow cytometry was used to determine the apoptosis rate and cell cycle. The shCtrl and shTROAP cell cultures were trypsinized and resuspended in DMEM after reaching the logarithmic growth phase. The cells were centrifuged at 1300 rpm for 5 min, the supernatant was discarded, and the cell precipitate was washed with D-Hanks (pH = 7.2–7.4) precooled at 4 °C. The cells were washed with 1 × binding buffer and centrifuged for 3 min. The pellets were resuspended in 200 μL 1 × binding buffer and stained with 10 μL Annexin V-APC for 10–15 min at 24° in the dark. The cells were analyzed using a flow cytometer (Thermo Fisher Scientific, Waltham, MA, USA) within 1 h, to determine the proportion of the cells in each cell cycle phase per group and the apoptosis rate. For the cell cycle assay, the cells were harvested in the same manner as for the apoptosis assay, washed, and precipitated. The harvested cells were centrifuged at 1300 rpm for 5 min and fixed in 75% ethanol precooled at 4 °C for at least 1 h. Next, the cells were centrifuged at 1300 rpm for 5 min to remove the fixant, and the cell precipitate was washed with D-Hanks. According to the number of cells, a certain volume of the cell staining solution (40 × PI mother solution (2 mg/mL): 100 × RNase mother solution (10 mg/mL): 1 × D-Hanks = 25: 10: 1000) (0.6–1 mL) was used, so that the cell passing rate was 300–800 cells/s when loading the machine. The cells were analyzed using a flow cytometer (Thermo Fisher Scientific, USA) within 1 h to determine the proportion of cells in each cell cycle phase per group. The data obtained were analyzed using the ModFit software (version 3.1). All assays were performed in triplicate.

### 4.7. Western Blot Analysis

Protein expression was analyzed via Western blotting. Briefly, the total protein was extracted using an immunoprecipitation protein lysis buffer (Beyotime Biotechnology, Shanghai, China). Protein concentrations were determined using the bicinchoninic acid assay (BCA) protein determination kit (Beyotime Biotechnology, Shanghai, China). Proteins were separated using sodium dodecyl sulfate-polyacrylamide gel electrophoresis (SDS-PAGE). Each protein sample (100 mg) was loaded onto 10% SDS-PAGE and transferred to a polyvinylidene fluoride (PVDF) membrane (Merckmillipore, Darmstadt, Germany) for 150 min. The primary antibodies were incubated in blocking buffer (TBST solution with 5% skim milk) overnight at 4 °C. The next day, the membranes were washed. However, the secondary antibodies were diluted with a blocking buffer, and the PVDF membranes were incubated for 1.5 h at room temperature and washed. Lastly, X-ray visualization was performed using the Pierce™ ECL Western blotting substrate kit (Thermo, USA). The following primary antibodies were used: anti-GAPDH (Santa Cruz, CA, USA), anti-TROAP (Sigma, Roedermark, Germany), anti-FLAG (Sigma-Aldrich, Germany), anti-STAT3 (CST, Danvers, DE, USA), anti-PLK1 (CST, USA), anti-RB1 (CST, USA), and anti-NPM1 (Abcam, Cambridge, UK). The secondary antibodies used included anti-Rabbit IgG (CST, USA) and anti-mouse IgG (CST, USA).

### 4.8. Wound-Healing Assay

The ShCtrl and shTROAP cells were cultured in DMEM supplemented with 10% fetal bovine serum. The cells were cultured in a 5% incubator at 37 °C, with three replicates per group, and the culture system at 100 μL/well. The next day, the low-concentration serum medium was changed, and a scratch meter was used to align the lower end of the 96-well plate at the central part; a scratch was formed by nudging upward. The cells were rinsed gently 2–3 times using a serum-free medium, and a low concentration of serum medium (0.5% fetal bovine serum (FBS)) was added. Pictures were taken at 0 h. The cells were cultured in a 37 °C, 5% CO_2_ incubator and scanned with Celigo at the appropriate time according to the degree of healing. The area of migration was analyzed using Celigo. This assay was performed in triplicate.

### 4.9. Cell Migration Assay and Invasion Assay

Cell migration was evaluated using a trans-well assay. The Corning transfer and invasion kits (Corning, NY, USA) were used for cell migration and invasion experiments. The invasion kit should be placed in a 37 °C incubator for 2 h, with 500 µL of a serum-free medium added to the upper and lower chambers to rehydrate the Matrigel matrix layer. The 786-O and ACHN cell lines were added to the kit at a density of 8 × 10^4^/well, 30% FBS medium was added to the lower chamber, and the cells were cultured for 24 h. The non-metastatic and non-invasive cells in the chamber were gently removed with a cotton swab. The chamber was placed in 4% paraformaldehyde fixative for 30 min and stained with Giemsa staining fluid (Sigma, Germany). The field of view was randomly selected for each trans-well chamber, and four (100×) and nine photos (200×) were taken. The 200× photos were used for cell counting and data analysis. This assay was performed in triplicate.

### 4.10. Tumor Formation Assay in a Nude Mouse Model

After trypsin digestion of the shCtrl and shTROAP tumorigenic cells in the logarithmic growth phase, the cell suspension was resuspended in the complete medium cell suspension (0.2 mL) containing 1 × 10^7^ cells injected into the right cutaneous axilla of 4-week-old BALB/c nude mice (Charles River, China). After 15 days, the tumor volumes were measured and collected 2 to 3 times per week for a total of 5 times. On day 33, an overdose of 2% sodium pentobarbital was administered, and death was confirmed via cervical dislocation. Lastly, the tumors were removed with medical scissors and tweezers, arranged on a whiteboard, and photographed for preservation. A ruler was required as a reference to read a specific scale. The animal experiment was reviewed and approved by the Ethics Review Committee of Jiangnan University (JN. No. 20201030c0561225[274]).

### 4.11. Co-Immunoprecipitation (Co-IP) Assay and Shotgun Mass Spectrometry Analysis

The experiment was performed using 293T cells. Using PCR, the gene sequence encoding the 3 × FLAG tag was fused to the 5’ end of the target gene to generate a 3 × FLAG-target fusion gene. Next, the 3 × FLAG-target fusion gene was inserted into the corresponding lentiviral expression vector to prepare the P3 × FLAG-Target expression plasmid. The empty lentivirus vector and P3 × FLAG-target plasmid were co-transfected into 293T cells with helper plasmids. Lentivirus particles, lenti-control, and lenti-3 × FLAG-target were harvested and prepared. The total cell protein in the supernatant was collected via centrifugation for BCA protein quantification. The protein complexes were purified by CO-IP using FLAG beads (Sigma). CO-IP samples were subjected to SDS-PAGE and Coomassie Brilliant Blue staining. According to each sample lane to be tested on the gel chart, the tape was cut off, and the protein in each sample tape was enzymatically hydrolyzed with trypsin, and the protein was cut into peptide segments. Liquid chromatography–tandem mass spectrometry (LC-MS) was performed for each peptide sample. Each sample’s original quality spectrum file was checked using the PD/MASCOT software (version 2.1) to obtain the protein identification results (Appendix A). Gene ontology and KEGG analyses were performed on the list of proteins specifically identified in the overexpression (OE) group. The genes that may interact with the target genes were selected for Western blot verification.

### 4.12. Statistical Analysis

Data were analyzed using the SPSS software (version 17.0). The Kaplan–Meier method was used to analyze the relationship between *TROAP* expression and prognosis, and a Cox regression model was used for multivariate analysis. The values are expressed as the mean ± SD. The Student’s *t*-test was used to analyze the differences between the two groups. Statistical significance was set at *p* < 0.05. The images were produced using the GraphPad Prism 8 and Illustrator CC 2018 software.

## 5. Conclusions

Our study confirmed that TROAP was involved in the proliferation and invasion of KIRC and was combined with STAT3 to exert a synergistic effect in promoting cancer, which is expected to be a new therapeutic target for KIRC.

## Figures and Tables

**Figure 1 ijms-24-09658-f001:**
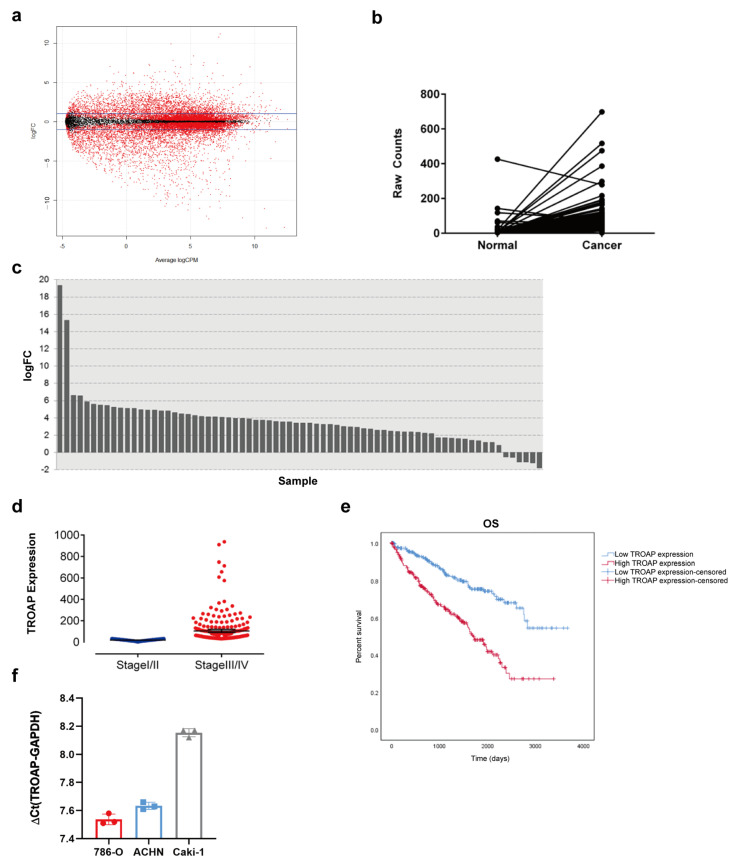
Difference expression of TROAP in KIRC. (**a**) A negative binomial general linear model was used to calculate the *p*-values, and the filtering criterion was <0.05. Simultaneously, the FC value of the sample was calculated using log2 (Cancer/Normal) and a filtering standard of ≥1 and ≤−1. (**b**,**c**) The difference in TROAP expression between the cancer and adjacent tissue samples in the TCGA database is displayed by FC (ratio of the expression level of cancer samples to adjacent samples) and *p*-value (statistical analysis model to judge whether it met the null hypothesis) in the form of a line chart and a bar chart. In the bar chart, the vertical axis is logFC (base 2 log value is taken for FC), the horizontal axis is different samples, and the logFC of each sample is represented by a bar. (**d**) TROAP expression in cancer tissues of patients with different pathological stages was different (*p* < 0.05). (**e**) The Kaplan–Meier method was used to analyze the effect of *TROAP* gene expression level on patients’ survival time. (**f**) *TROAP* mRNA expression abundance in 786-O, ACHN, and Caki-1 cells. In the cells with a relatively large ΔCt, the target gene expression abundance is relatively lower: when ΔCt value ≤ 12, the target gene expression abundance is high. ΔCt = Ct value of the target gene, i.e., the Ct value of the reference gene. Results are expressed as the mean ± standard error of the mean of the triplicate samples in (**f**). Circles, squares and triangles represented the corresponding ΔCt values of 786-O, ACHN and Caki-1, respectively.

**Figure 2 ijms-24-09658-f002:**
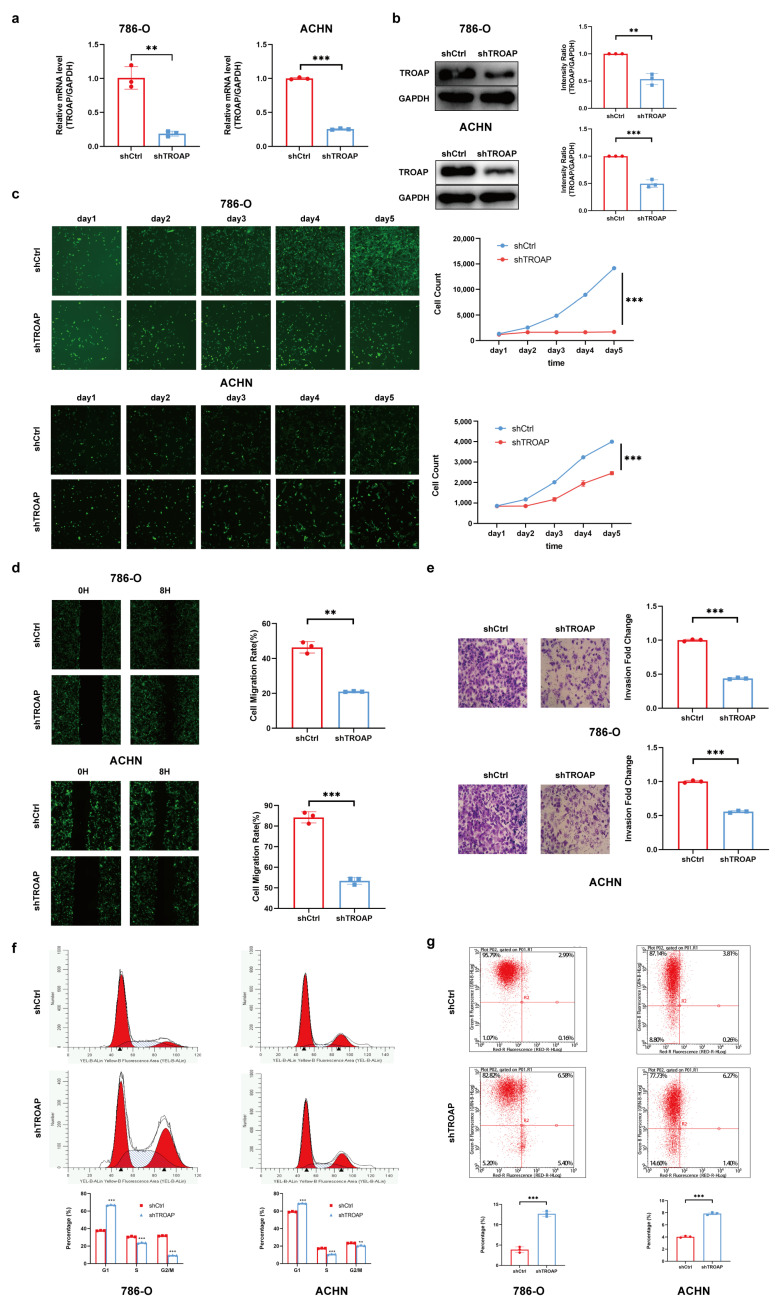
The function of TROAP in the 786-O and ACHN cell lines. (**a**) The shTROAP inhibition efficiency in the cell lines was determined by qRT-PCR. (**b**) Western blot was used to detect the expression of TROAP protein in shTROAP and shCtrl groups. (**c**) Cell proliferation in the shCtrl and shTROAP groups was assessed by Celigo assays. (**d**) Celigo scratch assay was used to investigate the migration of KIRC cells after TROAP inhibition. (**e**) The invasion of 786-O and ACHN cells via TROAP inhibition was investigated through a trans-well assay. The original magnification is 100×. (**f**) PI-FACS detected the effect of TROAP inhibition on the KIRC cell cycle. (**g**) The effect of TROAP inhibition on the apoptosis of KIRC cells was detected by Annexin V-APC single-staining. Results are expressed as the mean ± standard error of the mean of the triplicate samples. The circles and squares represent values for the groups shCtrl and shTROAP in (**a**,**b**,**d**,**e**,**g**). ** *p* < 0.01, *** *p* < 0.001.

**Figure 3 ijms-24-09658-f003:**
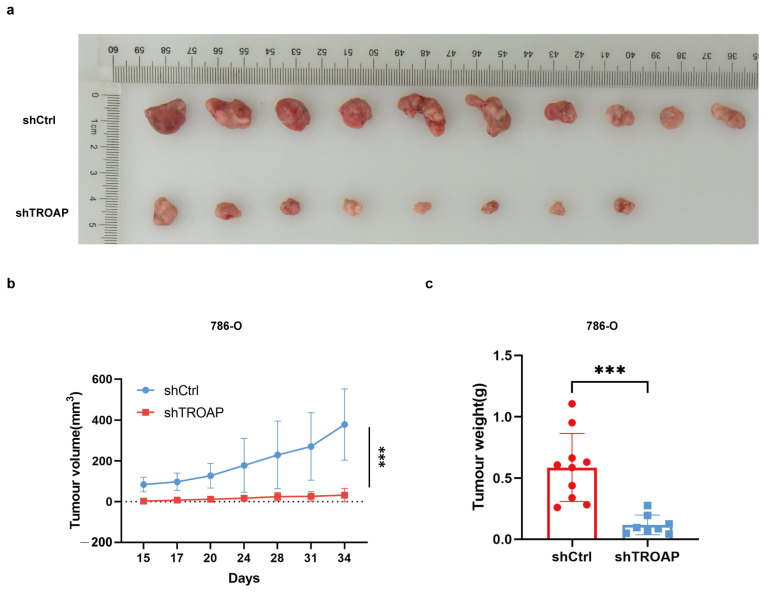
Low expression of *TROAP* inhibited tumor growth in BALB/c nude mice. (**a**) Images of the tumors obtained after resection 33 days after transplantation in nude mice. (**b**,**c**) Inhibition of *TROAP*-expression suppressed KIRC xenograft tumor growth, as measured by two-dimensional calipers of tumor volume and weight. Results are expressed as the mean ± standard error of the mean of triplicate samples. The circles and squares represent values for the groups shCtrl and shTROAP in (**c**). *** *p* < 0.001.

**Figure 4 ijms-24-09658-f004:**
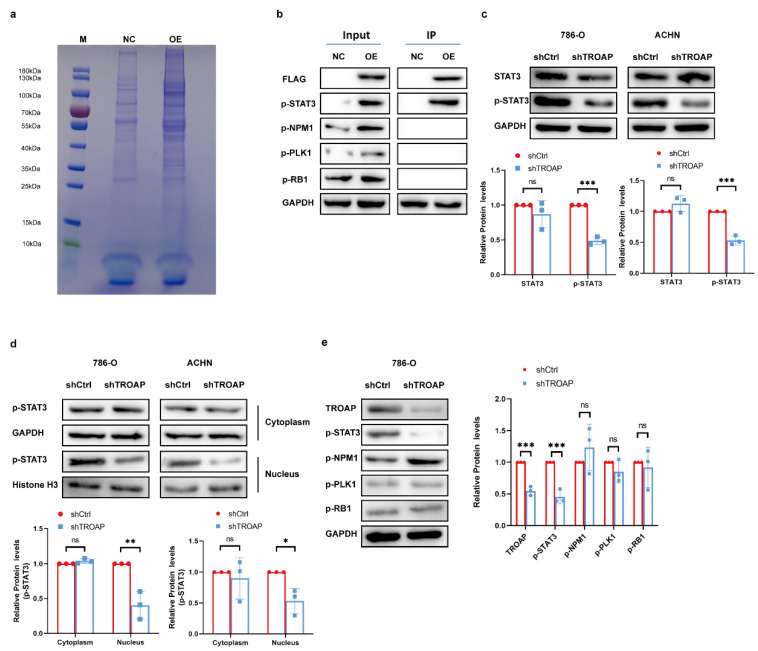
STAT3 is an interacting protein of TROAP. (**a**) SDS-PAGE electrophoresis and Coomassie brilliant blue staining of CO-IP samples. (**b**) The four best candidate proteins p-STAT3, p-PLK1, p-RB1 and p-NPM1 were selected for functional recovery experiments. Western blot results showed that p-STAT3could interact with TROAP. (**c**) Western blot analyzed p-STAT3 and total STAT3 protein levels in TROAP silencing cells. (**d**) p-STAT3 protein levels in subcellular fractions of KIRC cells with *TROAP* knockdown. GAPDH and Histone H3 were used as cytoplasmic and nuclear markers, respectively. (**e**) Western blot analysis of TROAP, p-STAT3, p-PLK1, p-RB1, and p-NPM1 in subcutaneous tumor tissues. Results are expressed as the mean ± standard error of the mean of the triplicate samples. Not significant (ns), * *p* < 0.05, ** *p* < 0.01, *** *p* < 0.001.

**Figure 5 ijms-24-09658-f005:**
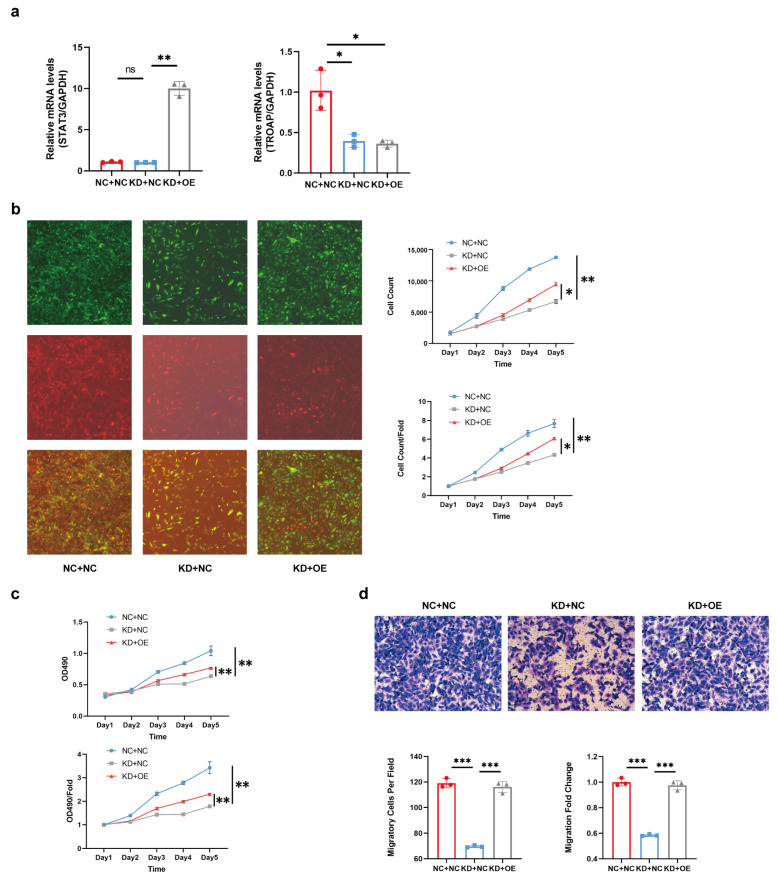
Functional restoration effect of STAT3 to TROAP inhibition. (**a**) qRT-PCR was used to detect each group’s mRNA expression of *TROAP* and *STAT3*. (**b**,**c**) The HCS proliferation screening analysis and MTT assay showed that the proliferation of the KD + NC group was significantly slower than that of the NC + NC group. Compared with the KD + NC group, the proliferation of the KD + OE group was significantly restored. (**d**) The trans-well assay showed that compared with the NC + NC group, the metastatic ability of the KD + NC group was weakened. Compared with the KD + NC group, the metastatic ability of the KD + OE group was enhanced. The original magnification is 100 ×. NC +NC: negative control virus-infected cell group; KD + NC: *TROAP*-knockdown + empty control virus infection cell group; KD + OE: *TROAP*-knockdown + STAT3 overexpression cell group. Results are expressed as the mean ± standard error of the mean of the triplicate samples. Circles, squares and triangles represented the corresponding values of groups NC+NC, KD+NC and KD+OE in (**a**,**d**). Not significant (ns), *p* ≥ 0.05, * *p* < 0.05, ** *p* < 0.01, *** *p* < 0.001.

**Figure 6 ijms-24-09658-f006:**
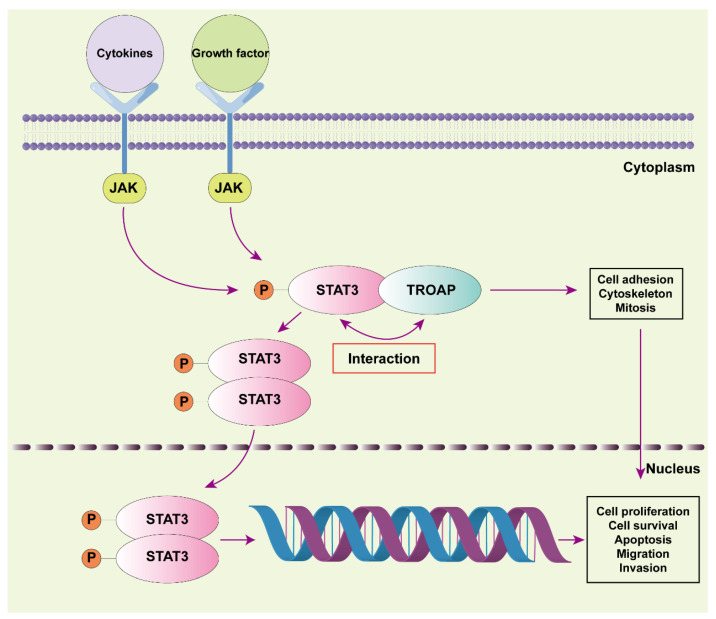
We propose a model for the interaction between TROAP and STAT3 in controlling KIRC cell proliferation, migration, and invasion. The JAK/STAT3 pathway is activated after the cytokine receptor receives growth factors and cytokine stimulation signals. Phosphorylation induces STAT3 protein dimerization, followed by nuclear translocation and DNA binding before it performs its nuclear function. In KIRC cells, we found that TROAP interacted with STAT3, which may affect this pathway and ultimately cell proliferation, metastasis, invasion, survival, and apoptosis. Numerous studies have shown that TROAP can also affect cell adhesion, cytoskeletal structure, and mitosis, thereby promoting tumor progression.

**Table 1 ijms-24-09658-t001:** Mann–Whitney U test for gene expression grouping.

		TROAP Expression Level		Total	*p*-Value
		High	Low		
TNM stage	T1/2	197	139	336	0.000
	T3/4	66	123	189	
Total		263	262	525	
Pathological stage	StageI/II	191	128	319	0.000
	StageIII/IV	72	134	206	
Total		263	262	525	

## Data Availability

The datasets generated during and/or analyzed during the current study are available from the corresponding author on reasonable request.

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
