# Peer review of "TROAP Promotes the Proliferation, Migration, and Metastasis of Kidney Renal Clear Cell Carcinoma with the Help of STAT3"

_ijms, 2023, doi:10.3390/ijms24119658_

Round 1

Reviewer 1 Report

The paper by Wang and coworkers investigates the role of trophinin-associated protein (TROAP) in kidney renal clear cell carcinoma (KIRC), describing TROAP expression, in vitro cell behaviour and the effects of TROAP inhibition in mice xenografts. Furthermore, TROAP regulation in vitro was also studied, shedding light on STAT3 involvement in KIRC progression.

The experimental design is clear and well presented. The materials and methods section, as well as the Results and the Discussion sections, are linear and well written.

However, the Introduction section should cite more works related to the role of TROAP in tumorigenesis, since other recent studies* reported those findings. Indeed, even if these studies are not specifically relevant to KIRC, they are useful to provide an updated state of art to the current manuscript.

* Liu H, Zhou Q, Xu X, Du Y, Wu J. ASPM and TROAP gene expression as potential malignant tumor markers. Ann Transl Med. 2022 May;10(10):586. doi: 10.21037/atm-22-1112. PMID: 35722431; PMCID: PMC9201116.

Reviewer 2 Report

The manuscript defined the expression of TROAP in KIRC cells and explored the role and downstream mechanism of TROAP in KIRC cells. The reviewer has the following concerns below.

1. Regarding the relationship between TROAP and cancers, there have been many reports so far, e.g. gastric cancer, colorectal cancer, and Breast Cancer……. While TROAP Promotes Breast Cancer (PMID: 31198787), colorectal cancer (PMID: 30021381), gastric cancer (PMID: 29956806), and renal clear cell carcinoma (PMID: 34287099 and the present study) Proliferation and Metastasis. What are the differences and advantages of existing reports? Is it not just replacing a tumor cell?

2. Figure 2f, 3b, 5b, and 5c lack statistical analysis.

3. Please provide the uncut image gel for all WB results.

4. The expression of TROAP, STAT3, p-STAT3, PLK1, RB1, and NPM1 and the levels of STAT3 and p-STAT3 in the cytoplasmic and nucleus should be detected in cancer tissues from BALB/c nude mice.

5. Figure 4, Poor picture quality. in special figure 4a, there are too many bands with no specific bands, how to determine the STAT3? In figure 4b, the reviewer cannot see any precise information. 

Round 2

Reviewer 2 Report

The authors have addressed my comments in round 1. I still have some minor comments that should be addressed.

1. All the figures have been revised to also reflect individual measurements rather than simple bars with SD.

2. The number of samples should be indicated in the Figure Legends.

3. Please statistically analyze the WB results in Figure 4.

4. Figure 4a is not convincing. What is the target? Compared to NC, there are too many strong bands, especially from 35 to 130 kDa.

5. Limitations of the study should be discussed.
